# Classification of a 3D Film Pattern Image Using the Optimal Height of the Histogram for Quality Inspection

**DOI:** 10.3390/jimaging9080156

**Published:** 2023-08-02

**Authors:** Jaeeun Lee, Hongseok Choi, Kyeongmin Yum, Jungwon Park, Jongnam Kim

**Affiliations:** 1Department of Artificial Intelligence Convergence, Pukyong National University, 45, Yongso-ro, Nam-gu, Busan 48513, Republic of Korea; jaeun0413@pknu.ac.kr (J.L.);; 2College of Business, Seoul National University, 1, Gwanak-ro, Gwanak-gu, Seoul 08826, Republic of Korea; 3Electronic and Computer Engineering Technology, University of Hawaii Maui College, 310 W Kaahumanu Ave, Kahului, HI 96732, USA; parkjung@hawaii.edu

**Keywords:** 3D film pattern image, classification, image processing, quality inspection, height of the histogram

## Abstract

A 3D film pattern image was recently developed for marketing purposes, and an inspection method is needed to evaluate the quality of the pattern for mass production. However, due to its recent development, there are limited methods to inspect the 3D film pattern. The good pattern in the 3D film has a clear outline and high contrast, while the bad pattern has a blurry outline and low contrast. Due to these characteristics, it is challenging to examine the quality of the 3D film pattern. In this paper, we propose a simple algorithm that classifies the 3D film pattern as either good or bad by using the height of the histograms. Despite its simplicity, the proposed method can accurately and quickly inspect the 3D film pattern. In the experimental results, the proposed method achieved 99.09% classification accuracy with a computation time of 6.64 s, demonstrating better performance than existing algorithms.

## 1. Introduction

A 3D film pattern image is a product designed to create a three-dimensional effect on printed patterns based on the amount and angle of light, as shown in Figure 1. Thus, although this film is 2D, it is a type of film where a 3D pattern image can be seen depending on factors such as the amount or angle of light. This product was recently developed for marketing purposes to attract consumer attention and favor. As shown in the product on the right side of Figure 2, the 3D film is applied to the surfaces of products such as cosmetics, liquor, and clothing. The 3D film is created by controlling the concentration of a 2D film, as shown in Figure 3a. After production, the 3D film is transferred to an inspection machine, as shown in Figure 3b, and then photographed using cameras. To capture the entire area, which is approximately 60 cm × 60 cm in size, four cameras are installed, and each camera captures a section divided into four regions. The films captured by the inspection device are then classified into either Figure 1a’s Good quality pattern images or Figure 1b’s Bad quality pattern images using the inspection algorithms. As mentioned above, the 3D film pattern image should exhibit a three-dimensional appearance. However, if a pattern image is of bad quality, as shown in Figure 1b, it does not create a 3D effect and cannot be sold. Therefore, pattern quality inspection is important. Furthermore, as the 3D film is a recently developed product, plans are underway to establish a mass production system, making it necessary to develop an algorithm to inspect the quality of the 3D film.

The good pattern of a 3D film is characterized by a clear pattern contour and high contrast, as shown in Figure 1a. On the other hand, a bad pattern has a vague pattern contour and low contrast, as shown in Figure 1b. These characteristics make it difficult to detect a defective 3D film during inspection as the bad pattern becomes invisible in three dimensions. Additionally, the size and texture of the 3D film pattern image are not uniform, posing challenges in the pattern examination. Furthermore, since it is a recently developed product, there are limited studies on the inspection techniques for it. Thus, there is a need to develop an inspection algorithm to identify defective 3D film products. Quality inspection for 3D films can be performed using existing research methods [1,2,3,4,5,6,7,8,9,10,11,12,13,14,15,16,17,18,19,20,21,22].

Since the 3D film pattern image was recently developed, there are few inspection methods to determine whether there is a defect. However, it is possible to consider a method of testing using existing methods. Among the existing methods, segmentation can be used as an inspection method based on the shape and texture of the 3D film. There are many techniques for segmentation, and binarization and segmentation are the most basic and used in many studies. Binarization changes pixels with high or low value to a black or white value based on a specific value. Binarization has several functions, and one of the representative methods is Otsu thresholding. Otsu thresholding is an algorithm that applies binarization by finding the optimal thresholding value via the brightness histogram of an input image. Although Otsu thresholding is stable, the computation is slow. As a result, the division accuracy can be low when strip detection is used as preprocessing. To compensate for this, Sa et al. proposed improved Otsu segmentation [1]. This method extracts grayscale information from the edge using the Sobel operator and then applies Otsu thresholding to extract grayscale information. This method showed better performance than classic Otsu thresholding by selecting appropriate thresholding. However, the paper presented a performance evaluation for the images with clear boundaries, such as scratch images. Therefore, it is considered challenging to extract objects from the background of the 3D film with bad patterns due to the vague contour.

Among the segmentation methods, the edge detection algorithm is an algorithm for edge detection in an image. In an image, the edge is the boundary between the foreground and background, and it is determined by the significant change in brightness or color intensity between the two regions. Therefore, the edge can be regarded as the part where the pixel intensity changes rapidly. Edge detection also has several techniques. Among them, Canny edge detection calculates the first-order differentiation for x and y and then differentiates it in four directions [2]. From the result of differentiation in four directions, the points with the maximum values become edges. The purpose of Canny edge detection is to identify strong edges, and it is less sensitive to noise than other edge detection algorithms. Recently, there were studies that combine Canny with other methods to improve edge detection accuracy. Pablo et al. proposed a method that obtains image information using Canny edge detection on images applied with HSV to increase edge detection accuracy in input images [3]. Mlyahilu et al. proposed a method for classifying 3D pattern images using a Convolutional Neural Network (CNN) [4]. In this method, the edges were first detected using preprocessing algorithms such as Canny, Sobel, and Laplacian, and then the images are classified as either good or defective based on the detected edges.

Recently, segmentation methods, such as morphological geodesic active contour methods, have been published [5,6,7]. Morphological geodesic active contour gradually develops segmentation areas and then morphologically segments the objects. This method combines the morphology snake algorithm and the geodesic active contour algorithm, which performs object detection using morphology operators [8,9,10,11,12,13]. Morphological geodesic active content has the advantage of being able to morphologically segment objects that are not in clear forms, such as squares and triangles. Recent medical studies have used these features to detect objects such as the lungs and cancer in CT images [5,6]. In addition, a welding bead inspection study using morphological geodesic active with preprocessing has been published to segment the uneven form of the welding bead [7]. However, it is cumbersome to set the location of the object to be partitioned and to find the optimal parameters value to obtain accurate results when using morphological geodesic active content. Therefore, it is not easy to get good results using this method in a factory where the experimental environment changes.

In addition, support vector machines (SVM) can be utilized as a method for classifying the presence or absence of defects in the 3D film [14,15,16,17,18]. Salman et al. published a study in which SVM was performed after applying Canny edge detection to identify and classify leaves [14]. Hsu and Chen proposed a study that detects blurry images and classifies them into categories [15]. However, finding the boundaries and optimal hyper-parameters for data classification in SVM still presents a challenge.

The Absolute difference and template matching can be used as methods to assess the similarity between 3D film patterns. Among them, the Absolute difference means the absolute value of image subtraction between two images [19]. If the two images are different, the ratio of the black area is high in the result of the Absolute difference. Among the methods using similarity, the template matching method is used to find a part in the image that matches the template image [20,21]. However, the patterns of 3D films analyzed in this paper vary in size and texture, and the shapes of good and bad patterns are similar to each other. Therefore, it is difficult to expect high accuracy when using template matching because of these features. In addition, a study using luminosity contrast methods was published to distinguish patterns in the image. Among these methods, the Michelson contrast is a technique that compares images using the maximum and minimum luminance values of the image [22]. The equation is as follows:(1)$$M_{n}=\frac{L_{max}-L_{min}}{L_{max}+L_{min}},$$
where $M_{n}$ represent the Michelson contrast value of the n-th input image, and $L_{min}$ and $L_{max}$ are the minimum and maximum luminance values in the input image. However, the Michelson contrast cannot guarantee high classification accuracy for 3D film pattern images. This is because Equation (1) only considers the minimum and maximum luminance values. In other words, even if the contrast is low, the bad pattern has several luminosity values at a certain level, as shown in Figure 1. Therefore, relying solely on the classification based on $L_{min}$ and $L_{max}$ may lead to a high probability of misclassification.

In this paper, we present an algorithm that can identify defective patterns in 3D film patterns by analyzing their image histogram from a statistical perspective. Our proposed algorithm utilizes the contrast features of 3D film pattern images to classify defects based on a specific height for the image histogram. The calculation process of the proposed algorithm is simple, but it achieves high accuracy in quality inspection and has low computational complexity. The remaining sections of this paper are organized as follows: Section 2 details our proposed algorithm, while Section 3 presents the experimental results obtained using this algorithm. Finally, in Section 4, we present the interpretation of the research findings, discuss the implications of the results, and draw conclusions from this study.

## 2. Materials and Methods

As described above, we can consider various methods for testing a 3D film pattern image using an existing algorithm. Among these methods, the segmentation method should be analyzed after detecting the pattern in the 3D film. However, it is challenging because the contour of the defective pattern is unclear, and the contrast is low. In addition, ensuring high accuracy in analysis using similarity methods such as SAD and template matching is not easy since all 3D film patterns do not have the same size and texture. Thus, the classification of 3D film pattern images is challenging. To solve these problems, we propose an inspection method to classify 3D film pattern images from a statistical perspective. The proposed algorithm utilizes a specific quantile in the image histogram to perform a quality inspection, and this method leverages the characteristics of the 3D film’s good and bad patterns. Despite its simplicity, the proposed method has the advantage of high accuracy with low computational complexity. In this section, we first explain how to cut each pattern in the 3D film in Figure 1 and then present how to classify using image histograms.

### 2.1. Fast Fourier Transform Algorithm for Cutting 3D Film Pattern Images

To inspect the pattern images in the 3D film of Figure 1, each image must be properly cropped according to the corresponding pattern. To accomplish this, we utilize the Fast Fourier transform algorithm proposed by Mlyahilu and Kim [23] to extract images by pattern in the 3D film. Let $I\left(i,j\right)$ be the N×N image in a spatial domain which is sampled from the Discrete Fourier Transform and is defined as $F\left(k,l\right)$, as shown in Equation (2).
(2)$$F\left(k,l\right)=\sum_{k=0}^{N-1}\sum_{l=0}^{N-1}I\left(i,j\right)e^{-2\pi t\left(\frac{ki}{N}+\frac{lj}{N}\right)},$$

For a color image $F\left(k,l\right)$, we convert to a gray image $I\left(n,m\right)$, as shown in Equation (3).
(3)$$I\left(n,m\right)=\frac{1}{N^{2}}\sum_{k=0}^{N-1}\sum_{l=0}^{N-1}F\left(k,l\right)e^{2\pi t\left(\frac{ka}{N}+\frac{lb}{N}\right)},$$

To obtain the horizontal $I_{a}\left(a,b\right)$ and vertical $I_{b}\left(a,b\right)$ indicated in Equations (4) and (5), respectively, we use the binaries $I\left(n,m\right)$ to determine the indices of white pixel values.
(4)$$I_{a}\left(a,b\right)=\left\{\begin{array}{cc} 255, & \;\;\;I\left(n,m\right)>0\\ \;\;\;\;0, & \;elsewhere \end{array}\right.$$
(5)$$I_{b}\left(a,b\right)=\left\{\begin{array}{cc} 255, & \;\;\;I\left(n,m\right)>0\\ \;\;\;\;0, & \;elsewhere \end{array}\right.,$$

By taking the intersection of the results from Equations (4) and (5), we obtain the pixel coordinates. Using this intersection information, we can then crop the pattern images from the 3D film.

### 2.2. Inspection Algorithm based on a Specific Height of Histogram

We classify the quality of the 3D film by following the procedure illustrated in Figure 4. Firstly, we input the 3D film as shown in Figure 1. The 3D film pattern image is cut using the Fast Fourier transform algorithm. At this point, the cut 3D film pattern images are inspected one by one. The image histogram is computed for the brightness values of the 3D film pattern image. A histogram is a method mainly used to deal with statistical data and is represented in the shape of a bar graph to understand the distribution of the data. In image processing, it is used to understand the pixel distribution of digital images. The image histograms provide information on the brightness and contrast of images, and various image processing techniques, such as histogram stretching, are applied to the analysis results of the histogram. Figure 5 shows the image histogram for each pattern of the 3D film: Figure 5a represents the good pattern image, and Figure 5b represents the bad pattern image. In addition, the image histogram expresses the *x*-axis as the pixel value and the *y*-axis as the frequency of the pixel. As shown in Figure 5a, the pixel value of the image histogram is distributed over a wide area since the pattern image has a clear outline and a high contrast. In contrast, the pixel values of the image histogram for the bad pattern image are concentrated in a specific range due to the blurred outline and low contrast of the image, as depicted in Figure 5b. Due to these characteristics, the image histogram height of the good pattern image is relatively lower than the image histogram height of the bad pattern image, as shown in Figure 5. The inspection algorithm proposed in this paper uses these characteristics to determine whether the 3D film pattern image is defective. Next, we find the pixel value that is the *x*-axis at the point intersecting the specific height α value of the tenth quartile in the histogram, as shown in Figure 6. After that, we find the height of the *y*-axis that matches the minimum value of the pixel value. This process aims to find the height ($Y_{\alpha }$). The reason why the minimum value is used is that the larger the pixel value in the image histogram of the 3D film pattern image, the more complicated the frequency value is, as shown in Figure 5. Therefore, the minimum value is used to accurately find the height of the point coinciding with the specific height α. The equation for calculating the height of the y-axis is as follows:(6)$$Y_{\alpha }=f(\underset{x}{\min}P_{\alpha }\left(x\right)),$$
where $P_{\alpha }\left(x\right)$ represents the pixel values x at the height α value of the image histogram in the gray image $I\left(n,m\right)$. In addition, $f\left(x\right)$ represents the height at the pixel value x of the image histogram. The decision can be defined as follows:(7)$$D=\left\{\begin{array}{cc} \;\;1, & \;\;\;Y_{\alpha }>\beta \\ 0, & \;otherwise \end{array}\right.,$$
where D is the class label of the image, and when $Y_{\alpha }>\beta$, it is a good image (D = 1), and when $Y_{\alpha }\leq \;\beta$, it is a bad image (D = 0).

## 3. Results

### 3.1. Data and Experimental Environment

An experiment was performed to evaluate the proposed algorithm using a total of 2850 3D film pattern images. Out of these 3D film pattern images, 2136 were classified as good pattern images, while 714 were classified as bad pattern images. Figure 7 shows some examples of the 3D film pattern images used in the experiment: Figure 7a shows the good pattern images, while Figure 7b shows the bad pattern images. The images in Figure 7 were obtained by applying the Fast Fourier transform algorithm to the 3D film in Figure 1, and each image had a size of 169 × 169. PC specifications were as follows: Window 10 Pro, Intel^®^ Core™ i7010700k CPU@3.80 GHz, 16 GB, NVIDIA GeForce RTX 2080 SUPER, and Python 3.11.

### 3.2. Evaluation Metrics

Accuracy is an indicator of the accurately classified ratio among the images, and sensitivity measures the ability of a test or model to correctly identify the positive cases. It is calculated as the ratio of the true positive results to the sum of the true positive and false negative results. On the other hand, specificity measures the ability of a test or model to correctly identify negative cases. It is calculated as the ratio of the true negative results to the sum of the true negative and false positive results. The calculation equations are as follows:(8)$$Accuracy=\left(TP+TN\right)/\left(TP+FP+FN+TN\right),$$
(9)$$Sensitivity=TP/\left(TP+FN\right),$$
(10)$$Specificity=TN/\left(TN+FP\right),$$
where the TP (True Positive) means the model was correctly classified in the positive class; FP (False Positive) means the model classified an observation to be positive when in reality it was actually negative; TN (True Negative) means the model correctly classified an observation in the negative class; and FN (False Negative) means the model incorrectly classified an observation as negative when it should have been classified as positive.

### 3.3. Performance of the Proposed Algorithm and Comparative Algorithms

A histogram was obtained for each cut image, as shown in Figure 5. In addition, the classification accuracy of each image histogram was calculated for different height intervals, ranging from 1/10 to 9/10, as shown in Table 1. The classification of the pattern images was obtained using Equation (7), with β set to 600. Table 1 shows the number of overlapping good and bad pattern images, enabling us to determine the number of incorrectly classified images. The accuracy, sensitivity, and specificity were all higher than 90% at all heights, as shown in Table 1. In addition, the highest results were obtained at 5/10, with an accuracy of 99.09% and a sensitivity of 99.91%. On the other hand, the lowest accuracy and specificity were shown at 1/10, and the bad pattern images were especially more misclassified than the good pattern images. As shown in Figure 5, this result was obtained because the width of the histogram was wide when both the good pattern and bad pattern images had a low height. In other words, the bad pattern has a characteristic where the frequency is high at a specific pixel value due to its low contrast. However, there may be other pixel values that occur less frequently in areas other than the specific pixel values. For this reason, the width at the low height of the image histogram has a relatively wide width compared to other heights.

To evaluate the performance of the proposed algorithm, we analyzed the results of the existing algorithms using the Abs-based difference method, Otsu thresholding, Canny edge detection, the CNN with Canny [4], morphological geodesic active contour [7], the Michelson contrast [22], the Canny with HSV [3], and the SVM with Canny [14]. For the CNN with Canny method proposed by Mlyahilu et al. [4], we used 32 and 64 nodes in the convolution layer, 10 epochs, the ReLu activation function, and Adam for the optimization function, as in the paper. In addition, the training set and test set were divided into 7:3, respectively. In addition, the SSIM was used to evaluate the similarity between the two images for the Absolute difference, Otsu thresholding, Canny edge detection, and morphological geodesic active control methods [24]. The SSIM evaluates the similarity using the structural information of the two images and compares the structure of the pixels constituting the image as shown in Equation (11).
(11)$$S\left(x,y\right)=b\left(x,y\right)\cdot c\left(x,y\right)\cdot r\left(x,y\right)\;\;\;\;\;=\frac{4\mu_{x}\mu_{y}\sigma_{xy}}{\left\{\left(\mu_{x}^{2}+\mu_{y}^{2}\right)\left(\sigma_{x}^{2}+\sigma_{y}^{2}\right)\right\}},\;\;\;\;\;where\;b\left(x,y\right)=\frac{2\mu_{x}\mu_{y}}{\mu_{x}^{2}+\mu_{y}^{2}},\;c\left(x,y\right)=\frac{2\sigma_{x}\sigma_{y}}{\sigma_{x}^{2}+\sigma_{y}^{2}},\;r\left(x,y\right)=\frac{2\sigma_{xy}}{\sigma_{x}\sigma_{y}}$$
where $b\left(x,y\right)$, $c\left(x,y\right)$, and $r\left(x,y\right)$ represent the average brightness, contrast, and correlation of the two images, respectively. $\mu_{x},$ $\mu_{y}$, $\sigma_{x}^{2}$, $\sigma_{y}^{2}$ and $\sigma_{xy}$ are the mean brightness of the original image, mean brightness of the image-processed image, the global variance of the original image, the global variance of the image-processed image, and the covariance between the original and image-processed images, respectively. Based on the results of the SSIM analysis, we concluded that the two images were considered similar if the resulting value was equal to or greater than the threshold value of 0.5. Conversely, if the resulting value was less than 0.5, we concluded that the two images were not similar. In the SSIM analysis, we utilized the first image of the good pattern in Figure 5a as the reference image. Based on the results of the SSIM analysis, we employed accuracy as the metric to evaluate the experimental results of the algorithms. Figure 8a displays the 3D film good pattern images, while Figure 8b to Figure 8f presents the results obtained from the Absolute difference, Otsu thresholding, Canny edge detection, Canny edge detection with HSV, and morphological geodesic active control algorithms for the 3D film good pattern images, respectively. In addition, Figure 9a displays the 3D film bad pattern images, while (Figure 9b) to Figure 9f displays the images obtained using the aforementioned comparative algorithms for the 3D film bad pattern images. The Absolute difference method represents the absolute value of the pixel difference between the good pattern image and other images, and the black area indicates a large difference. In all Canny Edge detections, the minimum threshold value and the maximum threshold value were set to 50 and 10, respectively. In addition, we marked a red line as a segmentation area for the morphological geodesic active content. Figure 8b is the result of obtaining the Absolute difference between the right pattern images, and the values of the two images are similar, so most of the white areas appear, but the black areas are distributed in the outline. Furthermore, in Figure 8a,c,f, a wider area than the actual pattern image was segmented, and in Figure 8d and Figure 8e, too many edges were detected, making it difficult to identify the pattern shape. In Figure 9a, it can be confirmed that the method cannot show a clear difference between the good and bad pattern images because the white and black areas are mixed in the outline of the pattern. In Figure 9c,f, fewer or more areas than the bad pattern images were detected, and the patterns were hardly detected in (Figure 9d) and (Figure 9e). Figure 10 and Table 2 present the confusion matrices and classification results for the good and bad 3D film images obtained using each respective existing algorithm and the proposed algorithm. As displayed in Table 2, the proposed algorithm had the highest performance with an accuracy of 99.09% when α = 5/10 and the computational time is 6.64 s. The Canny with the SVM method had an accuracy of 99.01%, but the calculation time was 5237.96 s, which took a very long time. In addition, both the Canny with HSV and morphological geodesic active content showed 74.5% accuracy. In addition, the sensitivity value is 0, indicating that they failed to classify any good patterns, as shown in Figure 10. Furthermore, CNN’s Canny algorithm exhibited very fast computation time, but showed lower accuracy, sensitivity, and specificity in the range of around 70%. All other methods had less than 50% accuracy. In particular, Canny with HSV and morphological geodesic active contour had a specificity value of 0, indicating that those algorithms failed to classify any bad pattern images, as shown in Figure 10. Thus, as observed in Figure 8 and Figure 9, the comparative algorithms failed to accurately detect the patterns, resulting in the inability to accurately classify the patterns, as shown in Table 2. Therefore, through the experimental results, we can confirm that the proposed method exhibits a superior performance compared to the comparative algorithms within a shorter period of time.

## 4. Discussion and Conclusions

The 3D film is a recently developed product, and there is a need for an inspection method to evaluate the quality of the 3D film pattern for mass production. However, there are few inspection methods since 3D films have only been recently developed. In addition, the good pattern of the 3D film has high contrast and a clear contour, while the bad pattern has low contrast and a cloudy contour, making it challenging to inspect 3D film images using conventional research. As a study for classifying the quality of 3D film, Mlyahilu et al. proposed a method for classifying 3D pattern images using the Convolutional Neural Network (CNN) [4]. In this method, the edges were first detected using preprocessing algorithms such as Canny, Sobel, and Laplacian, and then the images are classified as either good or defective based on the detected edges. In Figure 11a shows the 3D film bad pattern image used in the experiment and is classified as a bad pattern due to the white scratch on the image surface; Figure 11b shows the edge detection result for Canny; Figure 11c shows the edge detection result for Sobel; and Figure 11d shows the edge detection result for Laplacian. In order to classify a defect in the bad image, a scratch should be detected after performing edge detection. However, in all the results shown in Figure 11b–d, the pattern and scratch of the 3D film were not distinguished. Therefore, it can be confirmed that it is not easy to detect the pattern using edge detection methods in a pattern image.

In this paper, we propose a quality inspection algorithm for the 3D film pattern images. The proposed method is to compare and classify certain heights in histograms of 3D film pattern images. In addition, this method is simple, but it has the advantage of high accuracy and fast inspection time. In the experiment, we applied various methods for comparison purposes, including a segmentation method that can inspect the shape and texture of the 3D film, methods using pattern images, a method using luminance, and methods using machine learning. We then compared their performance with the proposed algorithm based on the results. In the experimental results, the proposed method shows an accuracy of 99.09% and the calculation time of 6.64 s. In contrast, all other comparison methods were less accurate than the proposed algorithm. Among the comparison methods, Canny with SVM showed the highest accuracy of 99.01%, but it took about 5237 s to calculate. Thus, it was judged to be impractical for use in real-time inspection due to the long processing time. However, our algorithm enables the company, which produces 3D film, to inspect the quality of the products within the required timeframe, ensuring both production speed and product quality, as shown in Table 1 and Table 2.

The idea proposed in this paper, which involves using the histogram of the image for inspection, may not be considered particularly groundbreaking. However, in conjunction with the aforementioned points, we have demonstrated in Table 2 that our method surpasses other approaches in terms of accuracy and processing time when inspecting images. Hence, the method proposed in this paper can be seen as a practical inspection approach that is conceptually straightforward, while still providing high accuracy and efficient processing time. Nevertheless, the proposed method has a limitation in that it cannot detect scratches or other defects present in the pattern image. To address this, future research plans include conducting a study to classify the defects mentioned above. Subsequently, the aim is to propose a system capable of performing real-time inspections immediately after the product is manufactured in the field.

## Figures and Tables

**Figure 1 jimaging-09-00156-f001:**
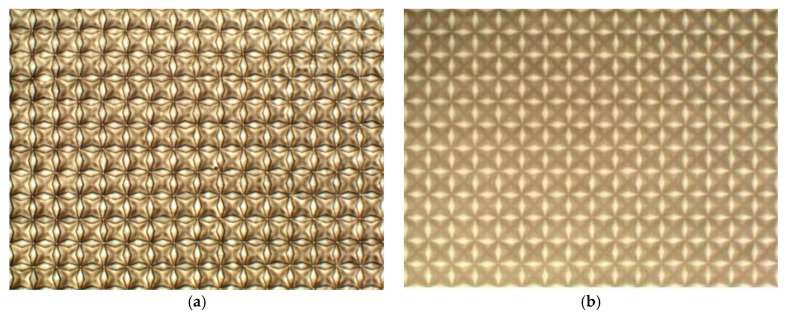
Two types of 3D film patterns. (**a**) Good pattern; (**b**) bad pattern.

**Figure 2 jimaging-09-00156-f002:**
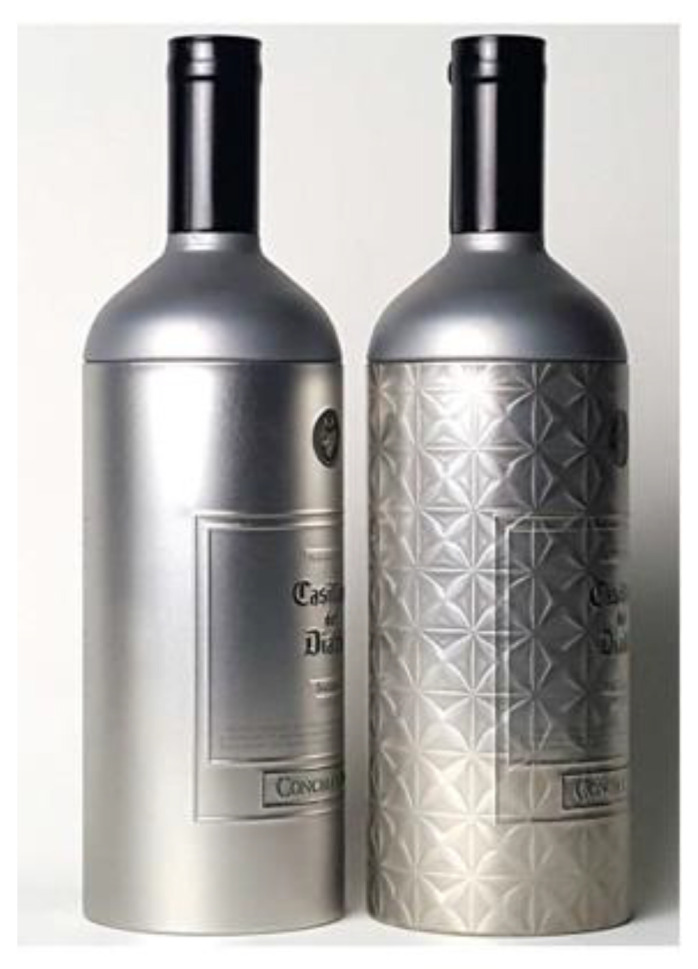
Production goods without and with 3D film good pattern images. (**Left**) Product without attached pattern image; (**Right**) product with attached pattern image.

**Figure 3 jimaging-09-00156-f003:**
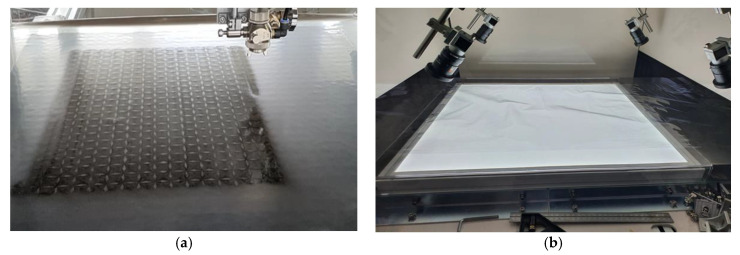
A 3D film pattern image manufacturing and inspection. (**a**) The 3D film manufacturing process; (**b**) inspection machine.

**Figure 4 jimaging-09-00156-f004:**
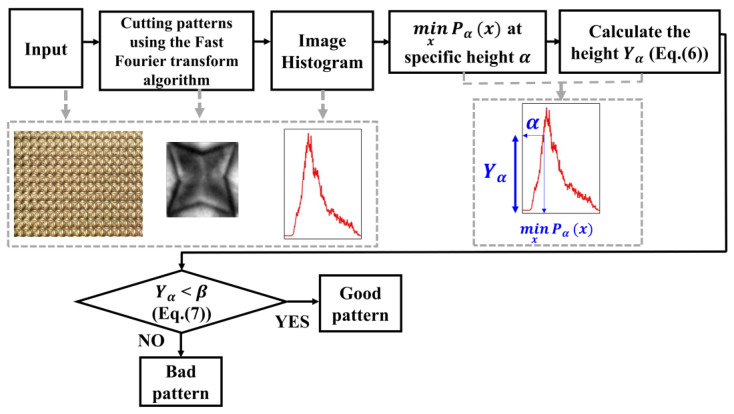
Procedures of the proposed algorithm.

**Figure 5 jimaging-09-00156-f005:**
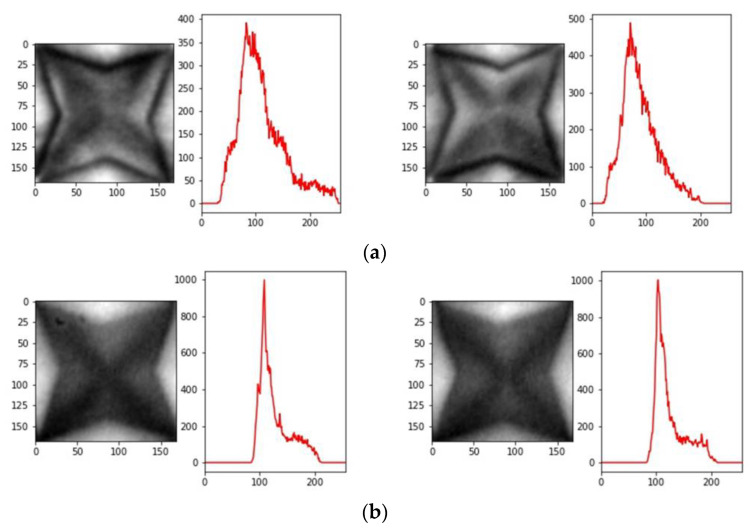
General 3D film pattern image with histogram. (**a**) Good pattern images; (**b**) bad pattern images.

**Figure 6 jimaging-09-00156-f006:**
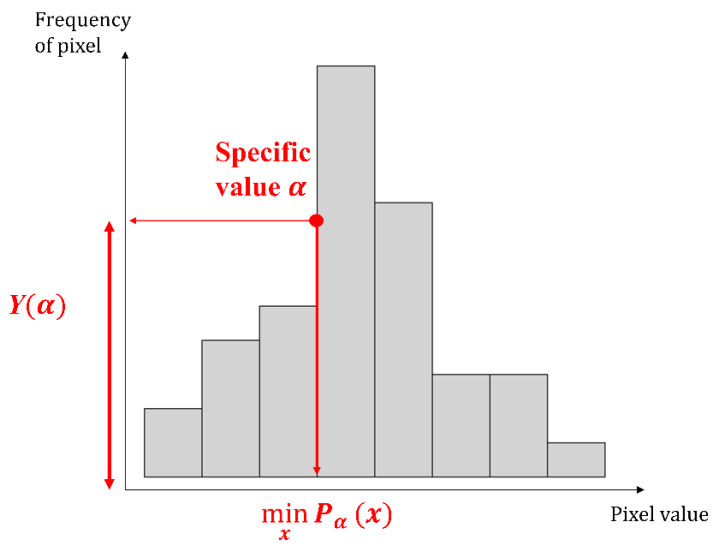
The height $Y_{\alpha }$ of the specific height α in the histogram.

**Figure 7 jimaging-09-00156-f007:**
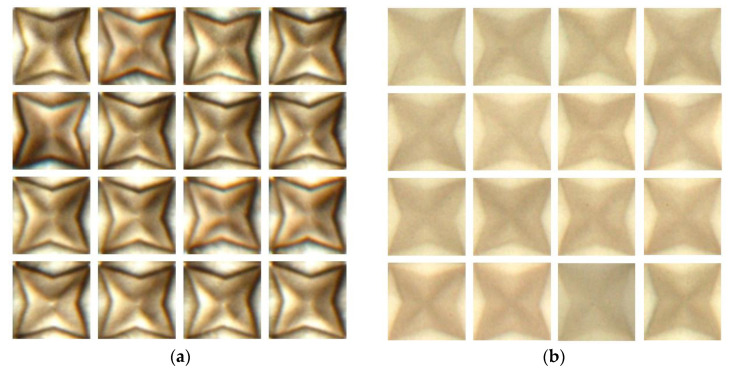
Cropped 3D film pattern images. (**a**) Good pattern images; (**b**) bad pattern images.

**Figure 8 jimaging-09-00156-f008:**
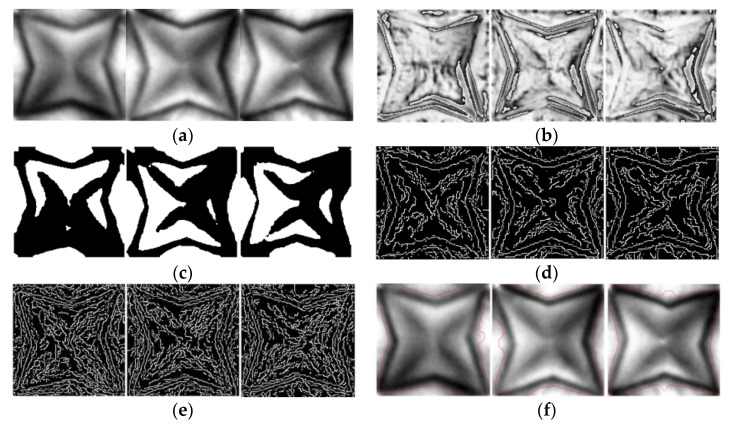
Good pattern images obtained using the various algorithms. (**a**) Original; (**b**) Absolute difference; (**c**) Otsu thresholding; (**d**) Canny edge detection; (**e**) Canny edge detection with HSV; and (**f**) morphological geodesic active contour.

**Figure 9 jimaging-09-00156-f009:**
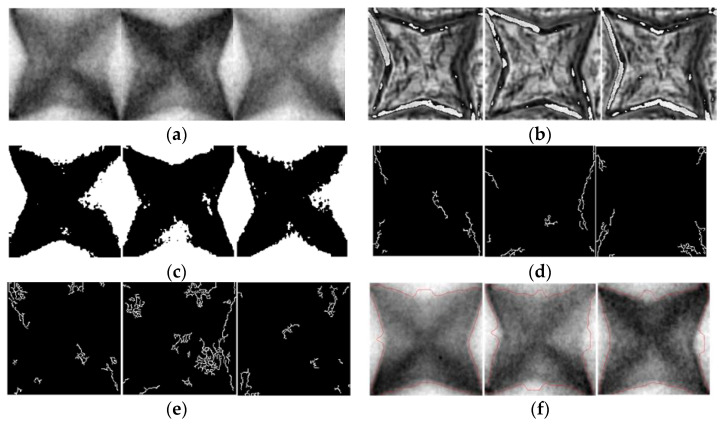
Bad pattern images obtained using the various algorithms. (**a**) Original; (**b**) Absolute difference; (**c**) Otsu thresholding; (**d**) Canny edge detection; (**e**) Canny edge detection with HSV; and (**f**) morphological geodesic active contour.

**Figure 10 jimaging-09-00156-f010:**
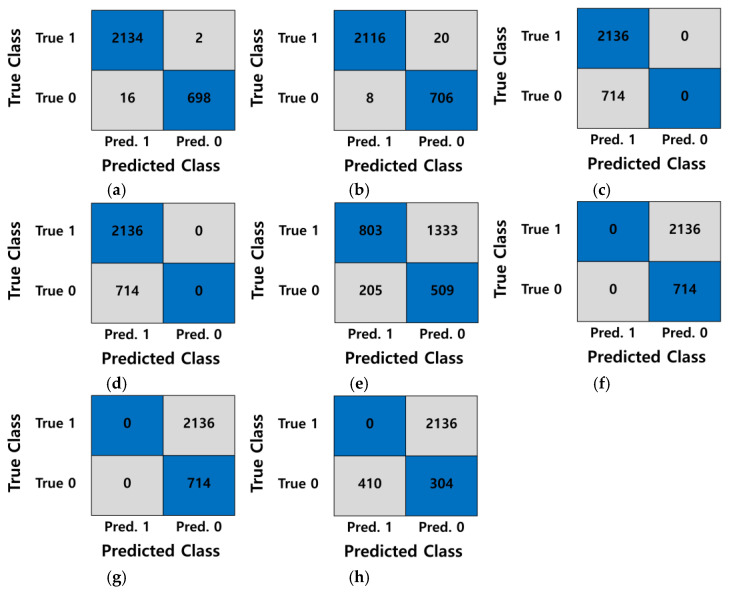
Confusion matrix. (**a**) Proposed algorithm (α=5/10); (**b**) SVM with Canny edge detection; (**c**) Canny edge detection with HSV; (**d**) morphological geodesic active contour; (**e**) Absolute difference; (**f**) improved Otsu thresholding; (**g**) Canny edge detection; and (**h**) Michelson Contrast.

**Figure 11 jimaging-09-00156-f011:**
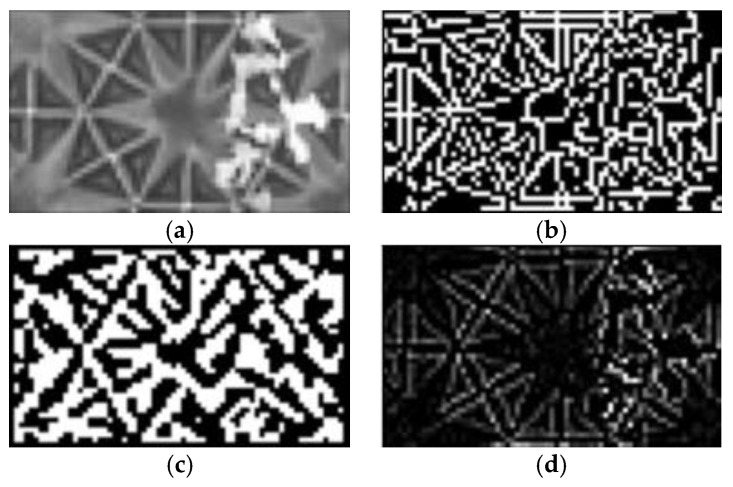
A 3D film bad pattern image and edge detection results from Canny, Sobel, and Laplacian methods. (**a**) Bad pattern image; (**b**) Canny; (**c**) Sobel; and (**d**) Laplacian.

**Table 1 jimaging-09-00156-t001:** Analysis of misclassification images and number of overlapping good pattern and bad pattern images, along with classification accuracy, sensitivity, and specificity of 3D film images.

	Height of Histogram	1/10	2/10	3/10	4/10	5/10	6/10	7/10	8/10	9/10

Misclassification images	Good pattern	108	145	54	35	2	26	44	12	34
	Bad pattern	132	20	24	15	16	11	14	15	24
Number of collapsed images		240	1653	78	50	18	37	58	27	58
Accuracy (%)		91.32	93.95	96.99	97.97	99.09	98.43	97.69	98.78	97.69
Sensitivity (%)		99.63	93.21	97.47	98.36	99.91	98.78	97.94	99.44	98.41
Specificity (%)		81.51	97.20	96.64	97.90	97.76	98.46	98.04	97.90	96.64

**Table 2 jimaging-09-00156-t002:** Classification results for algorithms with 3D film images.

		Accuracy (%)	Sensitivity	Specificity	Time (s)
Algorithm			(%)	(%)	
Proposed algorithm (α=5/10)		99.09	99.91	97.76	6.64
SVM with Canny [14]		99.01	99.06	98.88	5237.96
Canny with HSV [3]		74.95	100	0	55.75
Morphological geodesic active contour [7]		74.95	100	0	1931.13
CNN with Canny [4]		71.50	71.05	70.81	0.95
Absolute difference		46.04	37.59	71.29	5.76
Improved Otsu thresholding [1]		25.05	0	100	5.59
Canny edge detection		25.05	0	100	6.49
Michelson Contrast [22]		10.67	0	42.58	107.16

## Data Availability

Not applicable.

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
