# Peer review of "Classification of a 3D Film Pattern Image Using the Optimal Height of the Histogram for Quality Inspection"

_2313-433X, 2023, doi:10.3390/jimaging9080156_

Round 1

Reviewer 1 Report

The introduction doesn't provide references on previous works on the subject. However, part 2 "Related Works" does that in a convincing manner, this is why I consider that this is ok.

I would recommend to improve part 3 "Proposed algorithm" by clearly describing the novelty of the proposed method compared to other existing methods. The presentation of the research behind the proposed algorithm is quite general. The idea of using mainly image histograms should be properly explained, with more arguments.

The results are convincing and properly presented in comparison to other existing solutions.

I would also suggest to cut part 6 "Patents" if there is nothing to be mentioned.

Reviewer 2 Report

Summary: This paper presents a simple histogram-based method to classify the 3D fila patterns which can be utilized for quality inspection.  The method is simple and interesting, however there is no technical contributions to be published as a scientific journal article. Therefore, my recommendation is against considering this paper for publication. Here are few specific comments:

1.       It is claimed in the title that the work can help in quality inspection, but it is not clear from the method or results.

2.       Figure 2: it is not clear from the caption which one has with/without pattern. Is it one with bad and one with good pattern?

3.       Give more details to Figure 3 so that reader can understand how it can help in quality inspection. What are the sensors/cameras used for quality inspection. What is the metatrail under observation? How pattern quality helps in material quality?

4.       Figure 7 should be placed in the beginning of section 3. The main method should be started with reference to Figure 7.

5.       Table 2 shows some comparative results. However, it is not clear was the data same for all methods. How is the accuracy calculated? Why not to show visual results in the form of images from all the methods?

6.       Section 6. Patents should be removed.

7.       The list of references does not have some good venue papers.

Summary: This paper presents a simple histogram-based method to classify the 3D fila patterns which can be utilized for quality inspection.  The method is simple and interesting, however there is no technical contributions to be published as a scientific journal article. Therefore, my recommendation is against considering this paper for publication. Here are few specific comments:

1.       It is claimed in the title that the work can help in quality inspection, but it is not clear from the method or results.

2.       Figure 2: it is not clear from the caption which one has with/without pattern. Is it one with bad and one with good pattern?

3.       Give more details to Figure 3 so that reader can understand how it can help in quality inspection. What are the sensors/cameras used for quality inspection. What is the metatrail under observation? How pattern quality helps in material quality?

4.       Figure 7 should be placed in the beginning of section 3. The main method should be started with reference to Figure 7.

5.       Table 2 shows some comparative results. However, it is not clear was the data same for all methods. How is the accuracy calculated? Why not to show visual results in the form of images from all the methods?

6.       Section 6. Patents should be removed.

7.       The list of references does not have some good venue papers.

Reviewer 3 Report

This paper proposes a new method for inspecting the quality of 3D film patterns for mass production. The proposed method uses histograms to compare and classify certain heights in 3D film pattern images, making it simple yet accurate with fast inspection time. The authors compared the proposed algorithm to various other methods, including segmentation, pattern images, luminance, and machine learning, and found that the proposed algorithm showed the highest accuracy of 99.09% and took only 6.64 seconds to calculate. Overall, this paper presents a promising development in the inspection of 3D film patterns for mass production. But I have some questions to authors:

1)The article is not entirely clear on what parameter you create histograms. Pixel brightness?

2) In the examples of the "bad" patterns, an image with a violation of contrast was shown, but how does your algorithm deal with scratches and other defects? It would be nice to list the main defects encountered and perhaps how often they occur.

3) You have in some places separated text by pictures, it's better to fix it. (For example, line 275-284)

4) Line 286: "Fig ure 10"
